# Delayed Skip Connections for Music Content Driven Motion Generation

**Nelson Yalta**[*] **& Tetsuya Ogata**
Intermedia Art and Science Department
Waseda University
{nelson.yalta@ruri,ogata@}waseda.jp

**Kazuhiro Nakadai**
Honda Research Institute
Japan Co., Ltd.
nakadai@jp.honda-ri.com

## Abstract

In this study, we employ skip connections into a deep recurrent neural network for modeling basic dance steps using audio as input. Our model consists of two blocks, one encodes the audio input sequences, and another generates the motion. The encoder uses a configuration called convolutional, long short-term memory deep neural network (CLDNN) which handle the power features of audio. Furthermore, we implement skip connections between the contexts of music encoder and motion decoder (i.e. delayed skip) for consistent motion generation. The experimental results show that the trained model generate predictive basic dance steps from a narrow dataset with low error and maintains similar motion beat f-score to the baseline dancer.

## 1 Introducttion

Generating human motion is a current issue and an active research field in different areas. In computer graphics, allows to perform animated choreographies (Fukayama & Goto, 2015). Dancing, as a performing art, have several representations which complicate its modeling (Crnkovic-Friis & Crnkovic-Friis, 2016). Therefore, universal accepted dance steps, known as basic steps, allow to generate similar dance sequence for a musical genre. Music content driven motion generation (i.e. generating dance to the music) is a task with two main issues: motion as a time-series (Gan et al., 2015) and motion beat (Kim et al., 2003; Ho et al., 2013; Chu & Tsai, 2012) constraint.

Dance, due to its physical nature, is a nonlinear time-series data with large dimensionality information (Gan et al., 2015). To overcome this, a factored conditional restricted Boltzmann machine and recurrent neural network (Alemi et al., 2017) has the goal to map between the audio and motion features and generate a new dance sequence. Nevertheless, the dance generation requires higher computational capabilities or larger dataset and is constrained to the trained data. While dancing, significant changes of motion occurs at regular time length. This lapse is defined as motion beat and when dancing to the music, both music and motion beat should be synchronized. In prior works (Crnkovic-Friis & Crnkovic-Friis, 2016; Alemi et al., 2017), the motion required detailed information from the music track for generating it.

In this paper, we propose a deep learning model for generating basic dance steps to the rhythm of music, which handles and generates large motion sequences with high precision to its music beat. Inspired by Sainath et al. (2015a), we used a deep learning (DL) representation called convolutional, long short-term memory deep neural network (CLDNN), it has been widely used in speech processing tasks (Sainath et al., 2017; 2015b). The convolutional layers reduce the audio input dimensionality and the LSTM layers separated into two blocks (encoder-decoder) are trained to predict a next motion frame using as inputs the compressed audio features and motion frame from a previous frame. With this configuration, we handle larger dance sequence. To ensure the motion beat is constrained to the music beat, we propose delayed skip connections (Orhan, 2017; Wang, 2016) which forward the contexts from the music encoder into the motion decoder. That is, by combining the motion with music contexts, reduces the training time and increase the f-score of the motion beat. The proposed model shows improvement for the skip connection over a plain CLDNN encoder.

---

[*]The training and evaluation files of are available online.

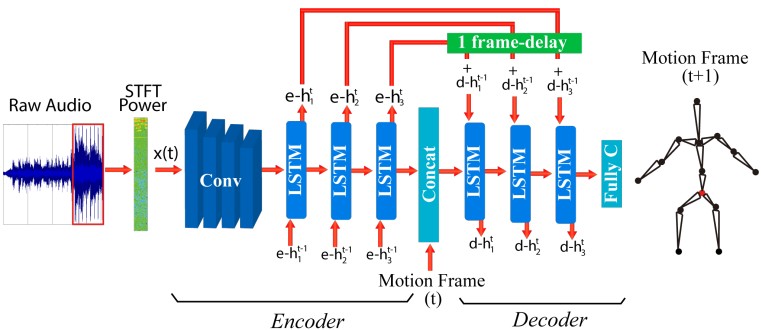

Figure 1: Proposed Framkework.

# 2 PROPOSED FRAMEWORK

## 2.1 DEEP RECURRENT NEURAL NETWORK

Mapping large dimensionality of sequences, such as motion, is challenging task for deep neural networks (DNN) (Gan et al., 2015). Furthermore, the highly nonlinear map between audio features and motion complicates this task (Alemi et al., 2017). For the construction of model, we consider a DRNN with LSTM layers separated in two blocks (Sutskever et al., 2014): one to reduce the music input sequence (encoder) and another for motion output sequence (decoder). This configuration allows to handle non-fixed dimensional signals such as motion and avoids that the performance decrease due to the long-term dependency of recurrent neural networks (RNN).

To handle the spectral variation of sound inputs, we implemented the encoder with the unified framework known as CLDNN (Sainath et al., 2015a). For motion generation tasks, CLDNN enhances the mapping between music and the motion. The proposed model is trained end-to-end with short sequence to generate motion. However, it requires larger training time or bigger dataset for generating a constrained motion from the training sequence.

We reduce the training time and enhance the motion beat f-score by implementing delayed skip connections into the model.

## 2.2 DELAYED SKIP CONNECTIONS

A wide study of skip connections is reported at Orhan (2017). A skip connection (i.e. shortcut) is an additional connection between different layers of a neural networks (He et al., 2015). This connection has been widely used in image processing (He et al., 2015; Hong et al., 2015) and audio processing (Kuleshov et al., 2017). For motion generation constrained to music, the decoder may suffer loose connection from the audio features. Therefore, in lower layers, the motion generation is constrained to the previous step or requires large training time to map the motion parameters with the music. Furthermore, a skip connection implemented between the output of the encoder layer and the input of the decoder may reduce the performance due to difference of the signal properties of music and motion.

To handle this, we implement skip connections (Fig. 1) between the contexts (i.e. delayed skip) of encoder and the decoder. This configuration ensures that the generated motion follows the music beat improving the motion beat f-score and reducing its training time.

# 3 RESULTS

We evaluate the performance of two models for generation motion using two metrics: prediction error and f-score. For the prediction error, we calculated the symmetric mean absolute percentage error which measures the distance between the dancer motion (True) and the motion generated by the models (Predicted). The f-score is employed to measure the motion beats generation w.r.t. the

Table 1: Salsa dataset results.

| Method | Prediction Error (%) | F-Score (%) |
|---|---|---|
| dancer | - | 47.50 |
| S2S | 17.83 | **58.07** |
| BSC-S2S | **17.58** | 50.83 |

Table 2: Prediction error (%) of dataset with tracks of different genres.

| Method | Bachata | Ballad | Bossa Nova | Rock | Hip Hop | Salsa |
|---|---|---|---|---|---|---|
| S2S (5 Epochs) | 18.89 | 19.49 | 20.37 | 17.06 | 17.56 | 20.38 |
| S2S (15 Epochs) | **14.50** | **14.30** | **15.00** | **13.58** | **13.05** | **16.02** |
| BSCS2S (5 Epochs) | 18.92 | 20.22 | 20.53 | 17.66 | 18.29 | 21.14 |
| BSCS2S (15 Epochs) | 14.96 | 14.89 | 15.41 | 13.95 | 13.52 | 16.34 |

Table 3: F-Score (%) of dataset with tracks of different genres.

| Method | Bachata | Ballad | Bossa Nova | Rock | Hip Hop | Salsa |
|---|---|---|---|---|---|---|
| Dancer (baseline) | 60.92 | 51.66 | 45.07 | 62.32 | 52.57 | 52.49 |
| S2S (5 Epochs) | 43.89 | 42.07 | 32.13 | 54.11 | **61.28** | 47.51 |
| S2S (15 Epochs) | 53.11 | 44.14 | 40.53 | **61.76** | 57.01 | 50.89 |
| BSCS2S (5 Epochs) | 51.30 | 46.49 | 40.39 | 49.50 | 60.87 | **52.49** |
| BSCS2S (15 Epochs) | **54.91** | **49.45** | **43.04** | 57.22 | 57.41 | 50.53 |

music beats. S2S is DRNN with CLDNN encoder without skip connections[1]. BSCS2S is S2S with skip connections. Each model is trained end-to-end for 5 epochs using a narrow dataset which contains 7 tracks of salsa music with 5 minutes each one. Each track contains two different basic dance steps which are permuted depending on the presence of voice. We also trained them with 13 tracks from 6 different genres. Each track also contains 1 or 2 basic steps per genre. For the last dataset, the models are trained for 5 and 15 epochs. Each training batch has 150 frame sequences.

Table 1 shows the prediction error and motion beat f-score for the salsa dataset. The table shows that S2S overcomes BSCS2S in f-score. S2S shows that when the training dataset contains few dance steps, the motion beat f-score can be improved after few epochs training.

Tables 2, 3 show the prediction error and motion beat f-score when the models are trained with tracks of different genres. The table 3 shows that BSCS2S overcomes S2S in most of the genres. This difference is maintained for larger training (15 epochs). BSCS2S handles betters when there is few information and multiple dance steps and improves the motion beat.

## 4 CONCLUSIONS

We proposed a method for motion generation using deep sequential learning which can be trained end-to-end. We showed that the models can generate correlated motion pattern with similar motion beat f-score to the dancer and lower prediction error. Also, due to the low forwarding time (approx. 12 milliseconds), they could be used for real time tasks. The models have a low training time and can be trained from the scratch. The proposed model shows reliable performance for motion generation. However, the motion pattern is affected by the diversity of the trained patterns and constrained to the dataset.

---

[1]Video samples can be found online

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
