# OpenReview forum: "DELAYED SKIP CONNECTIONS FOR MUSIC CONTENT DRIVEN MOTION GENERATION"
_ICLR.cc/2018/Workshop — Reject_

### Official Review · AnonReviewer3 · 2018-03-07
**limited in scope, inconclusive results**

**Rating:** 3
**Confidence:** 4

**Review:**

The paper describes a method for generating dance moves (in the form of motion capture data) from music audio, using a network architecture that makes use of convolutional layers as well as LSTMs. Two variants of the architecture are evaluated: one with so-called "delayed skip connections", and one without. Results are mixed, and which variant performs best seems to depend mostly on the choice of evaluation metric.

The writing could definitely be improved to be more clear. From reading the paper, it is hard to determine exactly what these delayed skip connections are, and how they modify the architecture. Several sentences in the paper stood out to me as vague or meaningless, or perhaps I simply didn't understand them in the way the authors intended:
* "Nevertheless, the dance generation requires higher computational capabilities or larger dataset and is constrained to the training data"
* "For motion generation tasks, CLDNN enhances the mapping between music and motion"
* "For motion generation constrained to music, the decoder may suffer loose connection from the audio features"

Given the length constraint, I think it would be best to get rid of such sentences that add no new information, and replace them with more details about what experiments were done exactly, and some motivation for the architectural choices that were made.

The main contribution of the paper is purported to be a comparison between a model with skip connections and one without, but at the same time, Orhan (2017) is cited as having conducted an extensive study on this topic already. So I'm not sure what kind of new information this work brings to the table.

Besides the presence of skip connections, the rest of the model architecture seems to be fixed for the experiments, but it is quite elaborate and many aspects of it are poorly motivated. This makes the comparison results less impactful as they may be specific to this particular architecture.

From tables 1, 2 and 3, my conclusion is that adding skip connections makes the prediction error worse, but improves the F-score. It is not clear from the paper which of these two metrics we should actually care more about. I'm also unsure about the statistical significance of the difference in prediction error in Table 1, given the small size of the dataset. Overall, these results seem very inconclusive and I feel that this is not adequately addressed in the text.

To summarise, given the limited scope, the poorly motivated architectural choices, the inconclusive results and the lack of novelty, I don't think this work is suitable as a workshop contribution.

---

### Official Review · AnonReviewer1 · 2018-03-09
**minimal contribution**

**Rating:** 3
**Confidence:** 3

**Review:**

This paper proposes to add skip connections into an encoder-decoder architecture to improve dance steps generation given a piece of music as input. The authors claim that adding the skip connections speed up training and improve the final results.

The paper is easy to follow, but many details are missing. For example, the network architecture, the size of the hidden layers, the training procedure are not specified. The data sets are also not clear from the description. The numbers in Table 1 also do not match those in Table 2 and 3. I assume they are from different epochs, but it has to be stated clearly.

Besides the missing details, the results are mixed in Table 2 and 3. I would not conclude that the proposed model is better. Some of the numbers actually go down after more epochs. The numbers also does not show what the authors claimed in previous sections.

Finally, adding skip connections is too weak of a contribution.

---

### Official Review · AnonReviewer4 · 2018-03-11
**unclear**

**Rating:** 4
**Confidence:** 4

**Review:**

The paper presents a neural network that predicts dance moves using music features as input. THis is generally an interesting problem, but it would be better to show that the proposed method works on some standard dataset, e.g. speech recognition or other sequence modeling tasks. As is, readers just cannot know if the proposed setup is reasonable, and what performance to expect.

I am confused by Figure 1: this is a sequence-to-sequence model? How so? All the activations are for time t (or t-1 and t+1) - so there is no sequence involved. Which activations get "concat"enated at the central layer? Does this maybe act more like a bottle-neck layer? Please clarify. Also, various kinds of skip and highwa connection schemes have been recorded in the litrature, th autors may want to take a look at these and discsus their model in light of this other body of work.

Finally, all the references for the CLDNN model that the authors present point to Google (Sainath's work). Maybe that is a sign that the paper is not as "widely spread" as the authors may want us to believe.

---

### Decision · Program_Chairs · 2018-03-20
**ICLR 2018 Workshop Acceptance Decision**

**Decision:**

Reject

**Comment:**

Based on the reviews, this paper has not been accepted for presentation at the ICLR workshop. However, the conversation and updates can continue to appear here on OpenReview.